# NEXT: A System for Real-World Development, Evaluation, and Application of Active Learning

**Kevin Jamieson**
UC Berkeley
kjamieson@berkeley.edu

**Lalit Jain, Chris Fernandez, Nick Glattard, Robert Nowak**
University of Wisconsin - Madison
{ljain,crfernandez,glattard,rdnowak}@wisc.edu

## Abstract

Active learning methods automatically adapt data collection by selecting the most informative samples in order to accelerate machine learning. Because of this, real-world testing and comparing active learning algorithms requires collecting new datasets (adaptively), rather than simply applying algorithms to benchmark datasets, as is the norm in (passive) machine learning research. To facilitate the development, testing and deployment of active learning for real applications, we have built an open-source software system for large-scale active learning research and experimentation. The system, called NEXT, provides a unique platform for real-world, reproducible active learning research. This paper details the challenges of building the system and demonstrates its capabilities with several experiments. The results show how experimentation can help expose strengths and weaknesses of active learning algorithms, in sometimes unexpected and enlightening ways.

## 1   Introduction

We use the term "active learning" to refer to algorithms that employ adaptive data collection in order to accelerate machine learning. By adaptive data collection we mean processes that automatically adjust, based on previously collected data, to collect the most useful data as quickly as possible. This broad notion of active learning includes multi-armed bandits, adaptive data collection in unsupervised learning (e.g. clustering, embedding, etc.), classification, regression, and sequential experimental design. Perhaps the most familiar example of active learning arises in the context of classification. There active learning algorithms select examples for labeling in a sequential, data-adaptive fashion, as opposed to passive learning algorithms based on preselected training data.

*The key to active learning is adaptive data collection.* Because of this, real-world testing and comparing active learning algorithms requires collecting new datasets (adaptively), rather than simply applying algorithms to benchmark datasets, as is the norm in (passive) machine learning research. In this adaptive paradigm, algorithm and network response time, human fatigue, the differing label quality of humans, and the lack of i.i.d. responses are all real-world concerns of implementing active learning algorithms. Due to many of these conditions being impossible to faithfully simulate active learning algorithms must be evaluated on real human participants.

*Adaptively collecting large-scale datasets can be difficult and time-consuming.* As a result, active learning has remained a largely theoretical research area, and practical algorithms and experiments are few and far between. Most experimental work in active learning with real-world data is simulated by letting the algorithm adaptively select a small number of labeled examples from a large labeled dataset. This requires a large, labeled data set to begin with, which limits the scope and scale of such experimental work. Also, it does not address the practical issue of deploying active learning algorithms and adaptive data collection for real applications.

To address these issues, we have built a software system called NEXT, which provides a unique platform for real-world, large-scale, reproducible active learning research, enabling

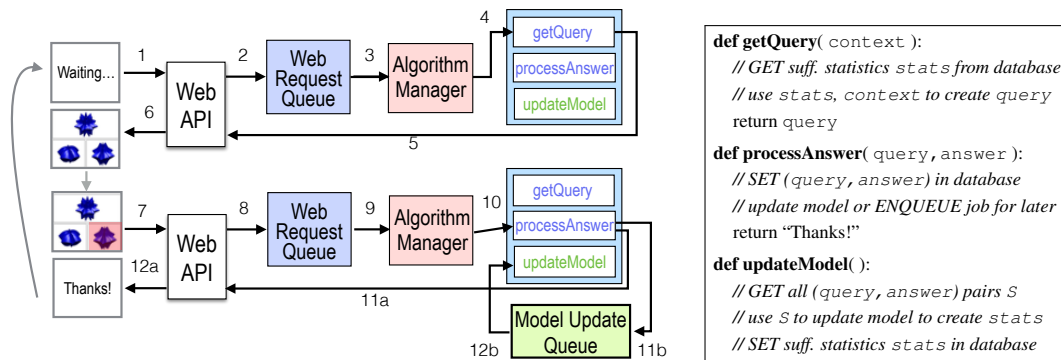

| 1. Request query display from web API | 8. Submit job to web-request queue |
|---|---|
| 2. Submit job to web-request queue | 9. Asynchronous worker accepts job |
| 3. Asynchronous worker accepts job | 10. Answer routed to proper algorithm |
| 4. Request routed to proper algorithm | 11a. Answer acknowledged to API |
| 5. Query generated, sent back to API | 12a. Answer acknowledged to Client |
| 6. Query display returned to client | 11b. Submit job to model-update queue |
| 7. Answer reported to web API | 12b. Synchronous worker accepts job |

Figure 1: NEXT Active Learning Data Flow

```
def getQuery( context ):
    // GET suff. statistics stats from database
    // use stats, context to create query
    return query
def processAnswer( query, answer ):
    // SET (query, answer) in database
    // update model or ENQUEUE job for later
    return "Thanks!"
def updateModel( ):
    // GET all (query, answer) pairs S
    // use S to update model to create stats
    // SET suff. statistics stats in database
```

Figure 2: Example algorithm prototype that active learning researcher implements. Each algorithm has access to the database and the ability to enqueue jobs that are executed in order, one at a time.

1. **machine learning researchers** to easily deploy and test new active learning algorithms;

2. **applied researchers** to employ active learning methods for real-world applications.

Many of today's state-of-the-art machine learning tools, such as kernel methods and deep learning, have developed and witnessed wide-scale adoption in practice because of both theoretical work and extensive experimentation, testing and evaluation on real-world datasets. Arguably, some of the deepest insights and greatest innovations have come through experimentation. The goal of NEXT is to enable the same sort of real-world experimentation for active learning. We anticipate this will lead to new understandings and breakthroughs, just as it has for passive learning.

This paper details the challenges of building active learning systems and our open-source solution, the NEXT system[1]. We also demonstrate the system's capabilities for real active learning experimentation. The results show how experimentation can help expose strengths and weaknesses of well-known active learning algorithms, in sometimes unexpected and enlightening ways.

## 2 What's NEXT?

At the heart of active learning is a process for sequentially and adaptively gathering data most informative to the learning task at hand as quickly as possible. At each step, an algorithm must decide what data to collect *next*. The data collection itself is often from human helpers who are asked to answer queries, label instances or inspect data. Crowdsourcing platforms such as Amazon's Mechanical Turk or Crowd Flower provide access to potentially thousands of users answering queries on-demand. Parallel data collection at a large scale imposes design and engineering challenges unique to active learning due to the continuous interaction between data collection and learning. Here we describe the main features and contributions of the NEXT system.

**System Functionality** - A data flow diagram for NEXT is presented in Figure 1. Consider an individual client among the crowd tasked with answering a series of classification questions. The client interacts with NEXT through a website which requests a new query to be presented from the NEXT web api. Tasked with potentially handling thousands of such requests simultaneously, the API will enqueue the query request to be processed by a worker pool (workers can be thought of as processes living on one or many machines pulling from the same queue). Once the job is accepted by a worker, it is routed through the worker's algorithm manager (described in the extensibility section below) and the algorithm then chooses a query based on previously collected data and sufficient statistics. The query is then sent back to the client through the API to be displayed.

After the client answers the query, the same initial process as above is repeated but this time the answer is routed to the 'processAnswer' endpoint of the algorithm. Since multiple users are getting

queries and reporting answers at the same time, there is a potential for two different workers to attempt to update the model, or the statistics used to generate new queries, at the same time, and potentially overwrite each other's work. A simple way NEXT avoids this race condition is to provide a locking queue to each algorithm so that when a worker accepts a job from this queue, the queue is locked until that job is finished. Hence, when the answer is reported to the algorithm, the 'processAnswer' code block may either update the model asynchronously itself, or submit a 'modelUpdate' job to this locking model update queue to process the answer later synchronously (see Section 4 for details). After processing the answer, the worker returns an acknowledgement response to the client.

Of this data flow, NEXT handles the API, enqueueing and scheduling of jobs, and algorithm management. The researcher interested in deploying their algorithm is responsible for implementing getQuery, processAnswer and updateModel. Figure 2 shows pseudo-code for the functions that must be implemented for each algorithm in the NEXT system; see the supplementary materials for an explicit example involving an active SVM classifier.

A key challenge here is latency. A getQuery request uses the current learned model to decide what queries to serve next. Humans will notice delays greater than roughly 400 ms. Therefore, it is imperative that the system can receive and process a response, update the model, and select the next query within 400 ms. Accounting for 100-200 ms in communication latency each way, the system must perform all necessary processing within 50-100 ms. While in some applications one can compute good queries offline and serve them as needed without further computation, other applications, such as contextual bandits for personalized content recommendation, require that the query depend on the context provided by the user (e.g. their cookies) and consequently, must be computed in real time.

**Realtime Computing** - Research in active learning focuses on reducing the *sample complexity* of the learning process (i.e., minimizing number of labeled and unlabeled examples needed to learn an accurate model) and sometimes addresses the issue of *computational complexity*. In the latter case, the focus is usually on polynomial-time algorithms, but not necessarily *realtime* algorithms. Practical active learning systems face a tradeoff between how frequently models are updated and how carefully new queries are selected. If the model is updated less frequently, then time can be spent on carefully selecting a batch of new queries. However, selecting in large batches may potentially reduce some of the gains afforded by active learning, since later queries will be based on old, stale information. Updating the model frequently may be possible, but then the time available for selecting queries may be very short, resulting in suboptimal selections and again potentially defeating the aim of active learning. Managing this tradeoff is the chief responsibility of the algorithm designer, but to make these design choices, the algorithm designer must be able to easily gauge the effects of different algorithmic choices. In the NEXT system, the tradeoff is explicitly managed by modifying when and how often the updateModel command is run and what it does. The system helps with making these decisions by providing extensive dashboards describing both the statistical and computational performance of the algorithms.

**Reproducible research** - Publishing data and software needed to reproduce experimental results is essential to scientific progress in all fields. Due to the adaptive nature of data collection in active learning experiments, it is not enough to simply publish data gathered in a previous experiment. For other researchers to recreate the experiment, the must be able to also reconstruct the exact adaptive process that was used to collect the data. This means that the complete system, including any web facing crowd sourcing tools, not just algorithm code and data, must be made publicly available and easy to use. By leveraging cloud computing, NEXT abstracts away the difficulties of building a data collection system and lets the researcher focus on active learning algorithm design. Any other researcher can replicate an experiment with just a few keystrokes in under one hour by just using the same experiment initialization parameters.

**Expert data collection for the non expert** - NEXT puts state-of-the-art active learning algorithms in the hands of non-experts interested in collecting data in more efficient ways. This includes psychologists, social scientists, biologists, security analysts and researchers in any other field in which large amounts of data is collected, sometimes at a large dollar cost and time expense. Choosing an appropriate active learning algorithm is perhaps an easier step for non-experts compared to data collection. While there exist excellent tools to help researchers perform relatively simple experiments on Mechanical Turk (e.g. PsiTurk [1] or AutoMan [2]), implementing active learning to collect data requires building a sophisticated system like the one described in this paper. To determine the needs

of potential users, the NEXT system was built in close collaboration with cognitive scientists at our home institution. They helped inform design decisions and provided us with participants to beta-test the system in a real-world environment. Indeed, the examples used in this paper were motivated by related studies developed by our collaborators in psychology.

NEXT is accessible through a REST Web API and can be easily deployed in the cloud with minimal knowledge and expertise using automated scripts. NEXT provides researchers a set of example templates and widgets that can be used as graphical user interfaces to collect data from participants (see supplementary materials for examples).

**Multiple Algorithms and Extensibility** - NEXT provides a platform for applications and algorithms. Applications are general active learning tasks, such as linear classification, and algorithms are particular implementations of that application (e.g., random sampling or uncertainty sampling with a C-SVM). Experiments involve one application type but they may involve several different algorithms, enabling the evaluation and comparison of different algorithms. The algorithm manager in Figure 1 is responsible for routing each query and reported answer to the algorithms involved in an experiment. For experiments involving multiple algorithms, this routing could be round-robin, randomized, or optimized in a more sophisticated manner. For example, it is possible to implement a multi-armed bandit algorithm inside the algorithm manager in order to select algorithms adaptively to minimize some notion of regret.

Each application defines an algorithm management module and a contract for the three functions of active learning: getQuery, processAnswer, and modelUpdate as described in Figure 2. Each algorithm implemented in NEXT will gain access to a locking synchronous queue for model updates, logging functionality, automated dashboards for performance statistics and timing, load balancing, and graphical user interfaces for participants. To implement a new algorithm, a developer must write the associated getQuery, processAnswer, and updateModel functions in Python (see examples in supplementary materials); the rest is handled automatically by NEXT. We hope this ease of use will encourage researchers to experiment with and compare new active learning algorithms. NEXT is hosted on Github and we urge users to push their local application and algorithm implementations to the repository.

## 3 Example Applications

NEXT is capable of hosting any active (or passive) learning application. To demonstrate the capabilities of the system, we look at two applications motivated by cognitive science studies. The collected raw data along with instructions to easily reproduce these examples, which can be used as templates to extend, are available on the NEXT project page.

### 3.1 Pure exploration for dueling bandits

The first experiment type we consider is a pure-exploration problem in the dueling bandits framework [3], based on the New Yorker Caption Contest[2] . Each week New Yorker readers are invited to submit captions for a cartoon, and a winner is picked from among these entries. We used a dataset from the contest for our experiments. Participants in our experiment are shown a cartoon along with two captions. Each participant's task is to pick the caption they think is the funnier of the two. This is repeated with many caption pairs and different participants. The objective of the learning algorithm is to determine which caption participants think is the funniest overall as quickly as possible (i.e., using as few comparative judgments as possible). In our experiments, we chose an arbitrary cartoon and $n = 25$ arbitrary captions from a curated set from the New Yorker dataset (the cartoon and all 25 captions can be found in the supplementary materials). The number of captions was limited to 25 primarily to keep the experimental dollar cost reasonable, but the NEXT system is capable of handling arbitrarily large numbers of captions (arms) and duels.

**Dueling Bandit Algorithms** - There are several notions of a "best" arm in the dueling bandit framework, including the Condorcet, Copeland, and Borda criteria. We focus on the Borda criterion in this experiment for two reasons. First, algorithms based on the Condorcet or Copeland criterion

| Caption | Plurality vote | Thompson | UCB | Successive Elim. | Random | Beat the Mean |
|---------|----------------|----------|-----|------------------|--------|---------------|
| My last of... | $0.215 \pm 0.013$ | $0.638 \pm 0.013$ | $0.645 \pm 0.017$ | $0.640 \pm 0.033$ | $0.638 \pm 0.031$ | $0.663 \pm 0.030$ |
| The last g... | $0.171 \pm 0.013$ | $0.632 \pm 0.017$ | $0.653 \pm 0.016$ | $0.665 \pm 0.033$ | $0.678 \pm 0.030$ | $0.657 \pm 0.031$ |
| The women'... | $0.151 \pm 0.013$ | $0.619 \pm 0.026$ | $0.608 \pm 0.023$ | $0.532 \pm 0.032$ | $0.519 \pm 0.030$ | $0.492 \pm 0.032$ |
| Do you eve... | $0.121 \pm 0.013$ | $0.587 \pm 0.027$ | $0.534 \pm 0.036$ | $0.600 \pm 0.030$ | $0.578 \pm 0.032$ | $0.653 \pm 0.033$ |
| I'm drowni... | $0.118 \pm 0.013$ | $0.617 \pm 0.018$ | $0.623 \pm 0.020$ | $0.588 \pm 0.032$ | $0.594 \pm 0.032$ | $0.667 \pm 0.031$ |
| Think of i... | $0.087 \pm 0.013$ | $0.564 \pm 0.031$ | $0.500 \pm 0.044$ | $0.595 \pm 0.032$ | $0.640 \pm 0.034$ | $0.618 \pm 0.033$ |
| They promi... | $0.075 \pm 0.013$ | $0.620 \pm 0.021$ | $0.623 \pm 0.021$ | $0.592 \pm 0.032$ | $0.613 \pm 0.029$ | $0.632 \pm 0.033$ |
| Want to ge... | $0.061 \pm 0.013$ | $0.418 \pm 0.061$ | $0.536 \pm 0.037$ | $0.566 \pm 0.031$ | $0.621 \pm 0.031$ | $0.482 \pm 0.032$ |

Table 1: Dueling bandit results for identifying the "funniest" caption for a New Yorker cartoon. Darker shading corresponds to an algorithm's rank-order of its predictions for the winner.

generally require sampling all $\binom{25}{2} = 300$ possible pairs of arms/captions multiple times [4, 3]. Algorithms based on the Borda criterion do not necessarily require such exhaustive sampling, making them more attractive for large-scale problems [5]. Second, one can reduce dueling bandits with the Borda criterion to the standard multi-armed bandit problem using a scheme known as the Borda Reduction (BR) [5], allowing one to use a number of well-known and tested bandit algorithms.

The algorithms considered in our experiment are: random uniform sampling with BR, Successive Elimination with BR [6], UCB with BR [7], Thompson Sampling with BR [8], and Beat the Mean [9] which was originally designed for identifying the Condorcet winner (see the supplementary materials for more implementation details).

**Experimental Setup and Results** - We posted 1250 NEXT tasks to Mechanical Turk each of which asked a unique participant to make 25 comparison judgements for \$0.15. For each comparative judgment, one of the five algorithms was chosen uniformly at random to select the caption pair and the participant's decision was used to update that algorithm only. Each algorithm ranked the captions in order of the empirical Borda score estimates, except the Beat the Mean algorithm which used its modified Borda score [9].To compare the quality of these results, we collected data in two different ways. First, we took union of the top-5 captions from each algorithm, resulting in 8 "top captions," and asked a different set of 1497 participants to vote for the funniest of these 8 (one vote per participant); we denote this the plurality vote ranking. The number of captions shown to each participant was limited to 8 for practical reasons (e.g., display, voting ease).

The results of the experiment are summarized in Table 1. Each row corresponds to one of the 8 top captions and the columns correspond to different algorithms. Each table entry is the Borda score estimated by the corresponding algorithm, followed by a bound on its standard deviation. The bound is based on a Bernoulli model for the responses and is simply $\sqrt{1/(4k)}$, where $k$ is the number judgments collected for the corresponding caption (which depends on the algorithm). The relative ranking of the scores is what is relevant here, but the uncertainties given an indication of each algorithm's certainty of these scores. In the table, each algorithm's best guess at the "funniest" captions are highlighted in decreasing order with darker to lighter shades.

Overall, the predicted captions of the algorithms, which generally optimize for the Borda criterion, appear to be in agreement with the result of the plurality vote. One thing that should be emphasized is that the uncertainty (standard deviation) of the top arm scores of Thompson Sampling and UCB is about half the uncertainty observed for the top three arms of the other methods, which suggests that these algorithms can provide confident answers with $1/4$ of the samples needed by other algorithms. This is the result of Thompson Sampling and UCB being more aggressive and adaptive early on, compared to the other methods, and therefore we recommend them for applications of this sort. **We conclude that Thompson Sampling and UCB perform best for this application and require significantly fewer samples than non-adaptive random sampling or other bandit algorithms.** The results of a replication of this study can be found in the supplementary materials, from which the same conclusions can be made.

## 3.2 Active Non-metric Multidimensional Scaling

Finding low-dimensional representations is a fundamental problem in machine learning. Non-metric multidimensional scaling (NMDS) is a classic technique that embeds a set of items into a low-

dimensional metric space so that distances in that space predict a given set of (non-metric) human judgments of the form "$k$ is closer to $i$ than $j$." This learning task is more formidable than the dueling bandits problem in a number of ways, providing an interesting contrast in terms of demands and tradeoffs in the system. First, NMDS involves triples of items, rather than pairs, posing greater challenges to scalability. Second, updating the embedding as new data are collected is much more computationally intensive, which makes managing the tradeoff between updating the embedding and carefully selecting new queries highly non-trivial.

Formally, the NMDS problem is defined as follows. Given a set $S$ comparative judgments and an embedding dimension $d \geq 1$, the ideal goal is to identify a set of points $\{x_1, \ldots, x_n\} \subset \mathbb{R}^d$ such that $||x_i - x_k||_2 < ||x_j - x_k||_2$ if "$k$ is closer to $i$ than $j$" is one of the given comparative judgments. In situations where no embedding exists that agrees with all of the judgments in $S$, the goal is to find an embedding that agrees on as many judgements as possible.

Active learning can be used to accelerate this learning process as follows. Once an embedding is found based on a subset of judgments, the relative locations of the objects, at least at a coarse level, are constrained. Consequently, many other judgments (not yet collected) can be predicted from the coarse embedding, while others are still highly uncertain. The goal of active learning algorithms in this setting is to adaptively select as few triplet queries (e.g., "is $k$ closer to $i$ or $j$?") as possible in order to identify the structure of the embedding.

**Active Sampling Algorithms** - NMDS is usually posed as an optimization problem. Note that

$$||x_i - x_k||_2 < ||x_j - x_k||_2 \iff x_i^T x_i - 2x_i^T x_k - x_j^T x_j + 2x_j^T x_k \iff \langle XX^T, H_{i,j,k} \rangle$$

where $X = (x_1, x_2, \ldots, x_n)^T \in \mathbb{R}^{n \times d}$ and $H_{i,j,k}$ is an all-zeros matrix except for the sub-matrix $\widetilde{H} = [1, 0, -1; 0, -1, 1; -1, 1, 0]$ defined on the indices $[i, j, k] \times [i, j, k]$. This suggests an optimization: $\min_{X \in \mathbb{R}^{n \times d}} \frac{1}{|S|} \sum_{(i,j,k) \in S} \ell \left( \langle XX^T, H_{i,j,k} \rangle \right)$, where $\ell$ in the literature has taken the form of hinge-loss, logistic-loss, or a general negative log-likelihood of a probabilistic model [10, 11, 12].

One may also recognize the similarity of this optimization problem with that of learning a linear classifier; here $XX^T$ plays the role a hyperplane and the $H_{i,j,k}$ matrices are labeled examples. Indeed, we apply active learning approaches developed for linear classifiers, like uncertainty sampling [13], to NMDS. Two active learning algorithms have been proposed in the past for this specific application [11, 14]. Here we consider four data collection methods, inspired by these past works: 1) (passive) uniform random sampling, 2) uncertainty sampling based off an embedding discovered by minimizing a hinge-loss objective 3) approximate maximum information gain sampling using the Crowd Kernel approach in [11], and 4) approximate maximum information gain sampling using the t-STE distribution in [12]. Care was taken in the implementations to make these algorithms perform as well as possible in a realtime environment, and we point the interested reader to the Supplementary Materials and source code for details.

**Experimental Setup and Results** - We have used NEXT for NMDS in many applications, including embedding faces, words, numbers and images. Here we focus on a particular set of synthetic 3d shape images that can be found in the supplementary materials. Each shape can be represented by a single parameter reflecting it's smoothness, so an accurate embedding should recover a one dimensional manifold. The dataset consists of $n = 30$ shapes selected uniformly from the 1d manifold, and so in total there are $30\binom{29}{2} = 12180$ unique triplet queries that could be asked. For all algorithms we set $d = 2$ for a 2d embedding (although we hope to see the intrinsic 1d manifold in the result). Each participant was asked to answer 50 triplet queries and 400 total participants contributed to the experiment. For each query, an algorithm was chosen uniformly at random from the union of the set of algorithms plus an additional random algorithm whose queries were used for the hold-out set. Consequently, each algorithm makes approximately 4000 queries.

We consider three different algorithms for generating embeddings from triplets, 1) embedding that minimizes hinge loss that we call "Hinge" [10], 2) the Crowd Kernel embedding with $\mu = 0.05$ that we call "CK" [11], and 3) and the t-STE embedding $\alpha = 1$ [12]. We wish to evaluate the sampling strategies, not the embedding strategies, so we apply each embedding strategy to each sampling procedure described above.

To evaluate the algorithms, we sort the collected triplets for each algorithm by timestamp, and then every 300 triplets we compute an embedding using that algorithm's answers and each strategy for

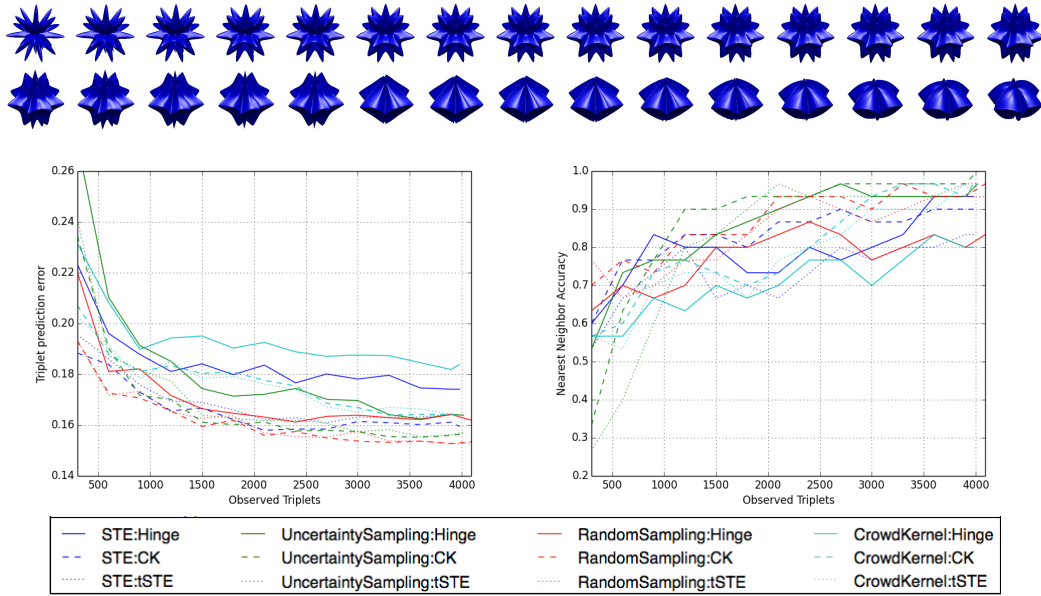

Figure 3: (Top) Stimuli used for the experiment. (Left) Triplet prediction error. (Right) Nearest neighbor prediction accuracy.

identifying an embedding. In Figure 10, the left panel evaluates the triplet prediction performance of the embeddings on the entire collected hold-out set of triplets while the right panel evaluates the nearest neighbor prediction of the algorithms. For each point in the embedding we look at its true nearest neighbor on the manifold in which the data was generated and we say an embedding accurately predicts the nearest neighbor if the true nearest neighbor is within the top three nearest neighbors of the point in the embedding, we then average over all points. Because this is just a single trial, the evaluation curves are quite rough. The results of a replication of this experiment can be found in the supplementary materials from which the same conclusions are made.

We conclude across both metrics and all sampling algorithms, the embeddings produced by minimizing hinge loss do not perform as well as those from Crowd Kernel or t-STE. In terms of predicting triplets, the experiment provides no evidence that the active selection of triplets provides any improvement over random selection. In terms of nearest neighbor prediction, UncertaintySampling may have a slight edge, but it is very difficult to make any conclusions with certainty. As in all deployed active learning studies, we can not rule out the possibility that it is our implementation that is responsible for the disappointment and not the algorithms themselves. However, we note that when simulating human responses with Bernoulli noise under similar load conditions, uncertainty sampling outperformed all other algorithms by a measurable margin leading us to believe that these active learning algorithms may not be robust to human feedback. **To sum up, in this application there is no evidence for gains from adaptive sampling, but Crowd Kernel and t-STE do appear to provide slightly better embeddings than the hinge loss optimization.** But we caution that this is but a single, possibly unrepresentative datapoint.

## 4 Implementation details of NEXT

The entire NEXT system was designed with machine learning researchers and practitioners in mind rather than engineers with deep systems background. NEXT is almost completely written in Python, but algorithms can be implemented in any programming language and wrapped in a python wrapper. We elected to use a variety of startup scripts and Dockerfor deployment to automate the provisioning process and minimize configuration issues. Details on specific software packages used can be found in the supplementary materials.

Many components of NEXT can be scaled to work in a distributed environment. For example, serving many (near) simultaneous 'getQuery' requests is straightforward; one can simply enlarge the pool of workers by launching additional slave machines and point them towards the web request

queue, just like typical web-apps are scaled. This approach to scaling active learning has been studied rigorously [15]. Processing tasks such as data fitting and selection can also be accelerated using standard distributed platforms (see next section). A more challenging scaling issue arises in the learning process. Active learning algorithms update models sequentially as data are collected and the models guide the selection of new data. Recall that this serial process is handled by a model update queue. When a worker accepts a job from the queue, the queue is locked until that job is finished. The processing times required for model fitting and data selection introduce latencies that may reduce possible speedups afforded by active learning compared to passive learning (since the rate of 'getQuery' requests could exceed the processing rate of the learning algorithm).

If the number of algorithms running in parallel outnumber the number of workers dedicated to serving the synchronous locking model update queues, performance can be improved by adding more slave machines, and thus workers, to process the queues. Simulating a load with stress-tests and inspecting the provided dashboards on NEXT of CPU, memory, queue size, model-staleness, etc. makes deciding the number of machines for an expected load a straightforward task. An algorithm in NEXT could also bypass the locking synchronous queue by employing asynchronous schemes like [16] directly in processAnswer. This could speed up processing through parallelization, but could reduce active learning speedups since workers may overwrite the previous work of others.

## 5 Related Work and Discussion

There have been some examples of deployed active learning with human feedback; for human perception [11, 17], interactive search and citation screening [18, 19], and in particular by research groups from industry, for web content personalization and contextual advertising [20, 21]. However, these remain special purpose implementations, while the proposed NEXT system provides a flexible and general-purpose active learning platform that is versatile enough to develop, test, and field any of these specific applications. Moreover, previous real-world deployments have been difficult to replicate. NEXT could have a profound effect on research reproducibility; it allows anyone to easily replicate past (and future) algorithm implementations, experiments, and applications.

There exist many sophisticated libraries and systems for performing machine learning at scale. Vowpal Wabbit [22], MLlib [23], Oryx [24] and GraphLab [25] are all excellent examples of state-of-the-art software systems designed to perform inference tasks like classification, regression, or clustering at enormous scale. Many of these systems are optimized for operating on a fixed, static dataset, making them incomparable to NEXT. But some, like Vowpal Wabbit have some active learning support. The difference between these systems and NEXT is that their goal was to design and implement the best possible algorithms for very specific tasks that will take the fullest advantage of each system's own capabilities. These systems provide great libraries of machine learning tools, whereas NEXT is an experimental platform to *develop, test, and compare* active learning algorithms and to allow practitioners to easily use active learning methods for *data collection*.

In the crowd-sourcing space there exist excellent tools like PsiTurk [1], AutoMan [2], and Crowd-Flower [26] that provide functionality to simplify various aspects of crowdsourcing, including automated task management and quality assurance controls. While successful in this aim, these crowd programming libraries do not incorporate the necessary infrastructure for deploying active learning across participants or adaptive data acquisition strategies. NEXT provides a unique platform for developing *active* crowdsourcing capabilities and may play a role in optimizing the use of human-computational resources like those discussed in [27].

Finally, while systems like Oryx [24] and Velox [28] that leverage Apache Spark are made for deployment on the web and model serving, they were designed for very specific types of models that limit their versatility and applicability. They were also built for an audience with a greater familiarity with systems and understandably prioritize computational performance over, for example, the human-time it might take a cognitive scientist or active learning theorist to figure out how to actively crowdsource a large human-subject study using Amazon's Mechanical Turk.

At the time of this submission, NEXT has been used to ask humans hundreds of thousands of actively selected queries in ongoing cognitive science studies. Working closely with cognitive scientists who relied on the system for their research helped us make NEXT predictable, reliable, easy to use and, we believe, ready for everyone.

## Footnotes

[1]The NEXT system is open source and available `https://github.com/nextml/NEXT`.

[2]We thank Bob Mankoff, cartoon editor of *The New Yorker*, for sharing the cartoon and caption data used in our experiments. www.newyorker.com

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
