[Reviews · NeurIPS 2015]

Submitted by Assigned_Reviewer_1

The paper presents an implementation to allow researchers to test active learning algorithms. They also evaluate two tasks using this framework.

The authors identify that the reproducibility of active learning experiments is a challenge because the labels requested varies by algorithm. However, given that the system obtains labels through crowd sourcing, how does the system ensure that subsequent executions get the same responses, or that compared systems receive the comparable quality crowd workers?

The paper would benefit from a review of literature in the hci community pertaining to the design of a crowd worker based active learning system, eg turkontrol.
Summary: The paper presents a practical implementation of an active learning system, but does not exhibit original research.

Submitted by Assigned_Reviewer_2

This paper is easy to follow, while there is not much novelty and the impact to the industry or academic is limited.
Summary: This paper introduces a system for active learning. The idea is not quite novel and the system seems not quite generalized to different applications.

Submitted by Assigned_Reviewer_3

This paper provides some high-level implementation details of a library for deploying and testing a variety of active learning schemes in a manner that is flexible enough to handle any number of experimental or real-world scenarios. This is an important contribution; while there existing libraries that provide implementations of a few active learning techniques in very limited scenarios, in general, a researcher or practitioner needs to build their own AL framework for their own task. The architecture described seems well-designed and scalable, in addition to being easily extendable.

The paper is well written and describes the contribution well. However, my main concern is the fit between the contribution (a practical description of new software) and the venue (typically a place where very theoretical machine learning is presented). To my knowledge, this is the first really flexible and scalable active learning library- the originality is above average- however, the difficulty of the described work isn't particularly high, the significance is only moderate.
Summary: Active learning researchers and practitioners will benefit greatly from the library described in this paper. However, aside from some mildly interesting interesting experimental results on the NMDS problem (eg, negative results that I personally always find valuable), there aren't really and direct contributions to the research. No new theorems or algorithms, instead a platform to host the output of newly developed results or algorithms.

Submitted by Assigned_Reviewer_4

This paper describes an open-source platform, NEXT, for deploying general active learning algorithms in a real-world setting, where users or clients provide live, real-time feedback through a web interface to queries formed by active learning algorithms. Aiming at facilitating both research in active learning itself and research in other fields that use active learning, the NEXT system hides away engineering details such as load-balancing, logging, communication, locking, etc and lets researchers focus on algorithm or experiment design. Two use cases are presented to illustrate the capability of the system in comparing different algorithms in a real active learning setting, as opposed to many existing studies that conduct comparisons in a simulation mode.

The system described in the paper is a great effort in facilitating active learning in the real world. Two suggestions on the presentation / doc side are as follows.

(1) While the two example applications in the paper are nice, it would be great if

there is also a template for a perhaps more typical active learning task, e.g., image or web page classification, that ML researchers/users can easily modify or extend.

(2) The documentation of NEXT seems to focus on launching the system in the cloud, but most users would probably want to start by running the system locally to get a feeling of how it works. It is not clear from the documentation how to do that.

Summary: This paper introduces an open-source system for applying active learning in the real world. It is a great effort in improving real-world experimentation and adoption of active learning. Two example applications and experiments demonstrate the capability of the system in comparing active learning algorithms in a live mode, as opposed to the many existing simulation studies.

Author Feedback
Author rebuttal: We thank the reviewers for their comments. We first clarify the contribution of our work over prior art for the first two reviewers and then address individual reviewer concerns.

Active learning in the setting we describe requires real-time computation to decide which question to ask next given all the data collected so far. The NEXT system reported in this paper is built to facilitate this and to make active learning experiments reproducible. To illustrate the need for such a system, it is instructive to consider a particular past case, the referenced Crowd Kernel paper. In that paper the authors state that they had to subsample and perform other approximations to run their algorithm in real-time. When we contacted the authors to get details in order to perform a faithful comparison, they could neither provide source code nor answer any detailed questions about the implementation. Thus, the specific algorithmic implementation and experimental conditions in which plots were presented in that paper cannot be reproduced. NEXT was designed to remedy this problem so that not only is the source code available, but if using cloud resources, one could come very close to perfectly recreating the algorithmic implementation and experimental conditions. For example, all the experiments we ran can be reproduced in minutes using the instructions on the NEXT Github page.

Reviewer_1 seems to misunderstand the novelty and purpose of NEXT (described above and thoroughly in the paper). The reviewer focuses on the issue of the quality and/or variability of crowd workers. This issue is relevant to crowd-based learning systems in general, but unrelated to the motivation and specific purposes of NEXT. Systems like Turkontrol are designed "to work around the variability in worker accuracy", and so they are unrelated but complementary to NEXT's capabilities.

Addressing Reviewer_6's concerns, though there are not yet any user studies we can cite, currently NEXT is being employed by two different psychology research groups to collect data from Mechanical Turk for cognitive science experiments. We look forward to their results.

Reviewer_4 suggested that NIPS may not be the ideal venue for this work. We submitted this work to NIPS after much thought and discussion with both theoretical and practical researchers. At the poster session at NIPS next December there will be many posters on multi-armed bandits and active learning research (mostly theoretical) motivated in part by minimizing human input, and almost none will have an experimental result with real (not simulated) human feedback. On the other hand, there will also be many posters describing algorithms justified sometimes by theorems but more often by their algorithm's performance on a number of real-world datasets (e.g. deep learning). While active learning is rightly guided by theory, we fear that there is a growing gap between theory and practice. Our goal is to make it easy enough to perform a real study so that we'll see more real-world experimental results on multi-armed bandits and active learning at future NIPS conferences. NEXT is a reasonable first step towards this direction.

As some reviewers point out, and as we admit in the paper, this paper is not a theoretical breakthrough or a game changing algorithm for active learning. It is the system that can lead to those things by inspiring theory from practice and discovering what truly works and what does not.